# Diagnostic test accuracy of simplified algorithms for diagnosing acute rheumatic fever: a systematic review

Rui Providencia [1,2] ✉, Ghazaleh Aali[3], Fang Zhu [4], Thomas Katairo[4], Mahmood Ahmad[1], Jonathan JH Bray[1], Ferruccio Pelone[1], Eloi Marijon[5], Miryan Cassandra[6], David S. Celermajer[7] & Farhad Shokraneh[1,3]

## Abstract

**Background** Rheumatic heart disease, the long-term sequel to acute rheumatic fever, remains a prevalent public health problem in Africa and other low to middle-income regions of the world. Diagnosing acute rheumatic fever and using the modified Jones criteria in high-prevalence areas remains challenging.
**Methods** We assessed the (i) diagnostic accuracy of simplified diagnostic algorithms among children, adolescents, and adults with suspected acute rheumatic fever, and (ii) the impact of different diagnostic criteria on the development of rheumatic heart disease (PROSPERO CRD42022344077). The MEDLINE, Embase, and Conference Proceedings Citation Index-Science were searched for relevant reports (date: 15th March 2025).
**Results** Here we identify 12,075 records, and three studies (four reports) meeting our eligibility criteria. Simplified diagnostic algorithms using only clinical data at community health centre-level (AUC 0.69, sensitivity 66% and specificity 68%), or adding 12-lead electrocardiogram and simple laboratory investigations at district-level facilities (AUC 0.76, sensitivity 77% and specificity 67%) perform worse than models including the full-set of laboratory investigations and echocardiography at National referral hospitals (AUC 0.91, sensitivity 84% & specificity 87%). Using modified Jones criteria without echocardiography results in an important loss of sensitivity (sensitivity 79%, specificity 100% & AUC 0.90). Progression to rheumatic heart disease is reported in 2.5–5% of children and young adults in high-prevalence areas who do not meet the full modified Jones criteria.
**Conclusions** Simplification of the modified Jones criteria in areas without access to echocardiography and laboratory investigations may lead to underdiagnosis of acute rheumatic fever. Some patients who do not meet the modified Jones criteria for definite acute rheumatic fever diagnosis may still progress to develop rheumatic heart disease.

## Plain language summary

Rheumatic heart disease, which can develop after acute rheumatic fever, is still common in parts of Africa and other low- to middle-income countries. Diagnosing acute rheumatic fever is difficult, especially in areas with reduced access to medical resources. Our review looked at how well simpler diagnostic tools work in such settings and found that basic methods using only clinical signs or simple tests miss many cases compared to full testing at major hospitals. Also, using the standard full testing misses some patients who later develop heart problems. More research is needed to find simple and reliable signs or tests that can help doctors spot the disease early—especially in areas where it's common—so that it doesn't lead to serious heart problems later on.

Acute rheumatic fever (ARF) is an inflammatory condition that may occur as a result of an autoimmune response to an upper respiratory tract infection caused by strains of group A beta-hemolytic streptococci. It occurs 10–21 days after the infection and may affect the heart (carditis), the main joints (arthritis), and sometimes the brain, skin, or subcutaneous tissues.

Rheumatic Heart Disease (RHD) is a condition characterized by structural and functional damage to the heart valves as a result of ARF[1]. Despite being a preventable disease, RHD, the long-term sequel to ARF, remains a prevalent public health problem in Sub-Saharan Africa, South and Southeast Asia, the Pacific Islands, and other low to middle-income regions of the world,

[1]GENEs health and social care evidence SYnthesiS unit, Institute of Health Informatics, University College, London, UK. [2]Barts Heart Centre, St Bartholomew's Hospital, Barts Health NHS Trust, London, UK. [3]Department of Evidence Synthesis, Systematic Review Consultants LTD, Oxford, UK. [4]Department of Biostatistics, Systematic Review Consultants LTD, Oxford, UK. [5]Paris Cardiovascular Research Centre, INSERM U970, European Georges Pompidou Hospital, Paris, France. [6]Cardiology Department, Hospital Dr. Ayres de Menezes, São Tomé, Sao Tome and Principe. [7]Faculty of Medicine and Health, The University of Sydney, Sydney, Australia. ✉e-mail: r.providencia@ucl.ac.uk

resulting in disability, low quality of life, early mortality, and a financial burden[2]. Globally, ARF has an incidence rate between 8 and 51 per 100,000[3]. It was estimated that there had been 319,400 deaths (95% CI, 297,300 to 337,300) in 2015 due to RHD. Furthermore, in the same year, there were 33.4 million (95% CI, 29.7–43.1 million) cases of RHD and 10.5 million (95% CI, 9.6–11.5 million) disability-adjusted life-years due to RHD globally[4]. In 2010, the cost of deaths due to RHD was estimated to be US$ 2200 billion (discounted) or US$ 5400 billion (undiscounted) globally[5].

The primary prevention of RHD is preventing the initial ARF attack, whereas secondary prevention is the protection from recurrent episodes of Group A Streptococcal infection and ARF through continuous antibiotic chemoprophylaxis. However, RHD can remain undetected for many years during the initial stages, thereby hindering the prophylaxis administration of penicillin[2]. Diagnosing ARF remains challenging. There is no single examination or test that can, in isolation, diagnose ARF. In 1944, Thomas Duckett Jones developed the main diagnostic tool. The original Jones criteria consisted of clinical aspects and laboratory findings[6] and underwent several modifications over the years. The American Heart Association published modified Jones criteria in 1956, 1965, 1984, and 1992[7]. The latest version of these criteria was published in 2015 and allowed diagnosing initial or recurrent ARF, made epidemiological considerations regarding the risk to contract the disease (different parameters for low vs moderate-high-risk populations), supported the use of echocardiography to assess for cardiac involvement (clinical and/or subclinical), and also included changes regarding arthritis and arthralgia[8,9]. Importantly, these criteria are complex, comprising multiple components, and often require laboratory tests and/or echocardiography, which may not always be accessible in low- and middle-income countries.

Depending on whether it is the first or a subsequent episode of the disease, a combination of minor and major criteria alongside the evidence of group A β-hemolytic streptococcal infection is needed[8]. Regardless of using Jones' criteria for diagnosing RF for decades, it is challenging for clinicians to avoid underdiagnosis or overdiagnosis of this condition. The main reasons are the lack of specific laboratory or clinical findings, especially in the early stage[10], and one-third of ARF cases reporting no prior sore throat[11]. Pharyngeal cultures can be associated with a considerable false-negative rate; therefore, serologic testing (as mentioned above) tends to be favored[12].

Advances in understanding the pathophysiology of ARF suggest that genetic susceptibility may play a role[13–15]. Previous analyses of twin studies showed a concordance rate of 44% in monozygotic twins and 12% in dizygotic twins, with the heritability of ARF estimated to be 60%[14]. Genetic variants within the human leukocyte antigen system and other immune processes have been identified[13]. Furthermore, the interest in biomarkers (i.e., Periostin, Tenascin-C, ischemia-modified albumin, Interferon-γ induced protein 10, IgG3-C4, etc.) has been growing[16–20], but convincing data on their validity for replacing the Jones criteria or to help improve their diagnostic accuracy is currently lacking.

The high prevalence of RHD in low- and middle-income countries that are under-equipped and understaffed necessitates using diagnostic criteria that are simple for experts and non-expert clinical staff to use. Current modified Jones criteria rely heavily on clinical manifestations, which can be subjective, transient, or hard to detect (i.e., arthritis is transient and carditis may be subclinical) and lead to over- and underdiagnosis. An accurate ARF diagnosis is important so antibiotic prophylaxis can be given to the right individuals and wrong diagnoses are avoided. This can prevent progression to RHD due to failure to diagnose ARF (i.e., false negatives) and wrong labeling of individuals as having ARF (false positives suffer the stigma of a diagnosis, which leads to some economic burden in areas known to be deprived).

We aim to systematically review the literature, identifying alternatives to the 2015 modified Jones criteria and assessing their diagnostic performance. Here, we show that simplified algorithms may lead to underdiagnosis, and some patients who do not meet the 2015 modified criteria for ARF may still develop RHD. The findings of this review help clarify that the assessed simplified algorithms cannot replace the standard diagnostic criteria. For individuals with a moderate to high risk of ARF and RHD, antibiotic treatment of suspected streptococcal pharyngitis may be of importance.

## Objective

To report the diagnostic accuracy of simplified algorithms compared to modified Jones criteria to identify ARF among children, adolescents, and adults with suspected ARF at health facilities.

## Methods

The Protocol was prospectively registered on PROSPERO (2022 July 4; CRD42022344077), and aimed to address the question raised by the World Health Organization Guideline Development Group for the WHO guideline on the prevention and diagnosis of rheumatic fever and rheumatic heart disease: *"Among children, adolescents and adults with suspected ARF, what is the diagnostic accuracy of simplified algorithms (compared to modified Jones criteria) to identify ARF at health facilities?"*

Search strategy/key words/databases

On 15th March 2025, we searched the following sources from the inception up to the search date:

- Embase via Ovid SP (1974–present)
- MEDLINE via Ovid SP (1946–present)
- Conference Proceedings Citation Index–Science (CPCI-S) (1990–Present)

Search strategies were developed by consulting the clinicians, controlled vocabularies (Medical Subject Headings=MeSH and Excerpta Medica Tree=Emtree), literature review, and test search results. Based on the recommendations from the 2nd edition of the Cochrane Handbook for Systematic Reviews of Diagnostic Test Accuracy[21], the searches were balanced between the sensitivity and specificity of the search results without applying a methodological search filter. Furthermore, the search was not limited to publication date, language, status, or document type.

The search strategies were peer-reviewed by another Information Specialist before the final run. The searches were run, documented, and reported by a senior information scientist and followed globally accepted guidelines: PRISMA 2020[22], PRISMA-S[23], PRISMA-DTA[24], PRISMA DTA for Abstracts[25], and PRESS[26].

The search strategies for all sources have been reported in Supplementary Table 1. We contacted the studies' authors as required to obtain the data or information.

## Inclusion/exclusion criteria

Population: children, adolescents, and adults with suspected ARF at healthcare facilities

Index test: simplified algorithms, including genetics, laboratory testing, imaging (echocardiography or other), and/or clinical variables (or reduced/simplified version of the Jones criteria) or other alternatives to provide an objective, more accurate, and quicker diagnosis of ARF will be considered.

Comparators: 1. Modified 2015 Jones criteria (Supplementary Tables 2–4).

## Other diagnostic criteria may have been used as comparators in eligible studies

Type of Studies: All diagnostic studies with usable data were included based on the Cochrane Handbook for Systematic Reviews of Diagnostic Test Accuracy[21], regardless of whether they were prospective or retrospective, pragmatic, or explanatory:

- Simple design studies with reference and index tests
- Studies with multiple groups of participants (including healthy controls)
- Studies with multiple reference tests
- Comparative test accuracy studies (randomized and non-randomized)

The primary outcome was ARF diagnosis, and the secondary outcome was the development of RHD following ARF.

Studies not assessing the diagnostic performance of a simplified or alternative diagnostic algorithm (i.e., not allowing assessment of index test performance vs reference), but providing information on the development of RHD following ARF using a simplified algorithm, were still considered eligible, and provided data for the secondary outcome analyses.

## Study selection

The search results were imported into EndNote 20, and the duplicate records were removed. The remaining records were then imported into Rayyan for double-blind screening by two reviewers. The blinding was inactivated when the screening was finished to resolve the conflicts.

## Data extraction

The following data were planned to be extracted from all included studies and double-checked by an independent reviewer.

- Study characteristics: authors, year of publication, country, study design, sample size, study period, setting, patient selection (random/ consecutive)
- Patient characteristics: patient type (age, sex, and subgroups), targeted condition (initial or recurrent ARF, etc.), number of patients examined by the test, number of patients who tested positive (false or true), follow-up period,
- Index test details: simplified algorithm definition, diagnostic criteria, etc.
- Reference test details: Jones criteria or other
- Outcomes: patients diagnosed with definite ARF and who developed RHD during follow-up.

## Quality assessment

The methodological quality of the included studies was planned to be assessed using the Quality Assessment of Diagnostic Accuracy Studies-2 (QUADAS-2) tool[27]. The QUADAS-2 checklist consists of four domains: (i) patient selection, (ii) index test, (iii) reference standard, and (iv) flow and timing, each of which is further divided into sub-items. Each domain was planned to be scored as 'yes' (positive assessment, high quality), 'no' (negative assessment, low quality), or 'unclear'. Disagreements between the two researchers were planned to be resolved by consensus or via a third party.

If the number of included studies in a meta-analysis was 10 or more, we planned to use the funnel plot to assess the publication bias based on the plot's symmetry.

The certainty of the evidence was planned to be rated using the Grading of Recommendations Assessment, Development and Evaluation (GRADE) methodology for diagnostic tests[28–30]. We planned to use GRADEpro to create this table for the diagnostic question. The five domains (risk of bias, indirectness, inconsistency, imprecision, and publication bias) were set to be judged as without concerns, with serious concerns, or with very serious concerns. The assessment of the certainty of the evidence was considered to be high when studies were cross-sectional. The reason for each of the five domains was planned to be judged as not serious, serious (downgraded by one level), or very serious (downgraded by two levels).

## Data synthesis, including meta-analyses

An overview of the available studies and demographics of patients was planned to be summarized for the included studies. The risk of bias assessment using QUADAS-2 was planned to be summarized in a table and/ or graph using Review Manager 5.4[31].

For the meta-analysis, we planned to summarize diagnostic accuracy statistics (sensitivity and specificity, PPV/NPV, LR+/LR−) with their 95% confidence intervals by using a bivariate random-effects model through Meta-DiSc 2.0[32]. Where possible, a summary receiver operating characteristic summary was fitted as described by the Cochrane Collaboration[21], and we assessed the area under the curve (AUC) using the R-package mada[33].

Data on the prevalence of ARF in children, adolescents, and adults with suspected disease presenting at healthcare facilities following the implementation of the modified Jones criteria[8] and the utilization of echocardiography are sparse. We utilized data from our combined cohort of two studies to estimate this prevalence (25.5%; 230 out of 903), and used it for estimating the effects per 1000 patients with suspected ARF assessed with the different simplified diagnostic approaches identified in our search.

Heterogeneity was planned to be assessed via visual inspection of the forest plot.

## Subgroup analysis

Depending on the availability of the data, we planned the following subgroup analyses:

- Subgroup by populations: young children <5 years, older children 5–9 years, young adolescents 10–14 years, older adolescents 15–19 years, adults 20 years and above, and pregnant women
- Subgroup by reference tests used: Modified Jones criteria vs other diagnostic criteria

## Reporting summary

Further information on research design is available in the Nature Portfolio Reporting Summary linked to this article.

# Results

## Search results and study selection

Our broad search identified 12,075 records, and we screened 9272 unique records. We found ten records possibly relevant during the record screening phase, but seven were excluded after more detailed assessment, and two were abstracts where further data is required before making a final decision (Fig. 1). More information on the excluded studies can be obtained from Supplementary Table 5.

Three studies, comprising a total of four records[34–37], met the inclusion criteria for our systematic review. Two studies, one in Uganda[34] and one in Sudan[35], provided data for the assessment of the diagnostic test accuracy of 5 simplified or modified diagnostic algorithms for ARF among individuals with a suspected diagnosis. The Uganda cohort had subsequent follow-up[36] and the outcome for individuals classified as definite and possible ARF was assessed, as well as for those with an alternate diagnosis. A third study[37], utilizing data from the Northern Australia RHD register, did not have a diagnostic test accuracy study design but provided data on the outcome of patients with definite, probable, and possible ARF, and rates of subsequent definite ARF and RHD. Data on study design and patient population are provided in Table 1.

## Study appraisal

The methodological quality of Ndagire et al.[34] and Ali et al.[35] was assessed using the QUADAS-2 tool (Supplemental Data 1). With regards to the use of GRADE to assess certainty of evidence, we had no concerns regarding indirectness or imprecision. However, we could not assess inconsistency or publication bias, as each of the different assessed simplified/modified diagnostic approaches was published in only one study.

## Diagnosis of ARF

Ndagire and colleagues[34] used cross-sectional data from patients enrolled from three districts in Uganda to conduct a predictive modeling study, utilizing data from three types of healthcare settings (community, district hospitals, and a tertiary hospital). The authors developed three different models, with increasing levels of complexity, for diagnosing ARF. Out of the 503 individuals, there were 143 definitive ARF cases, and a simplified diagnostic algorithm using only clinical data (from history-taking and clinical examination) obtained at community health centers displayed a moderate discriminative capacity (AUC 0.69, sensitivity 66% and specificity 68%) when compared to the modified Jones criteria (Table 2). A subsequent model adding 12-lead ECG and simple laboratory investigations performed at district-level facilities had an improved performance (AUC 0.76) at the

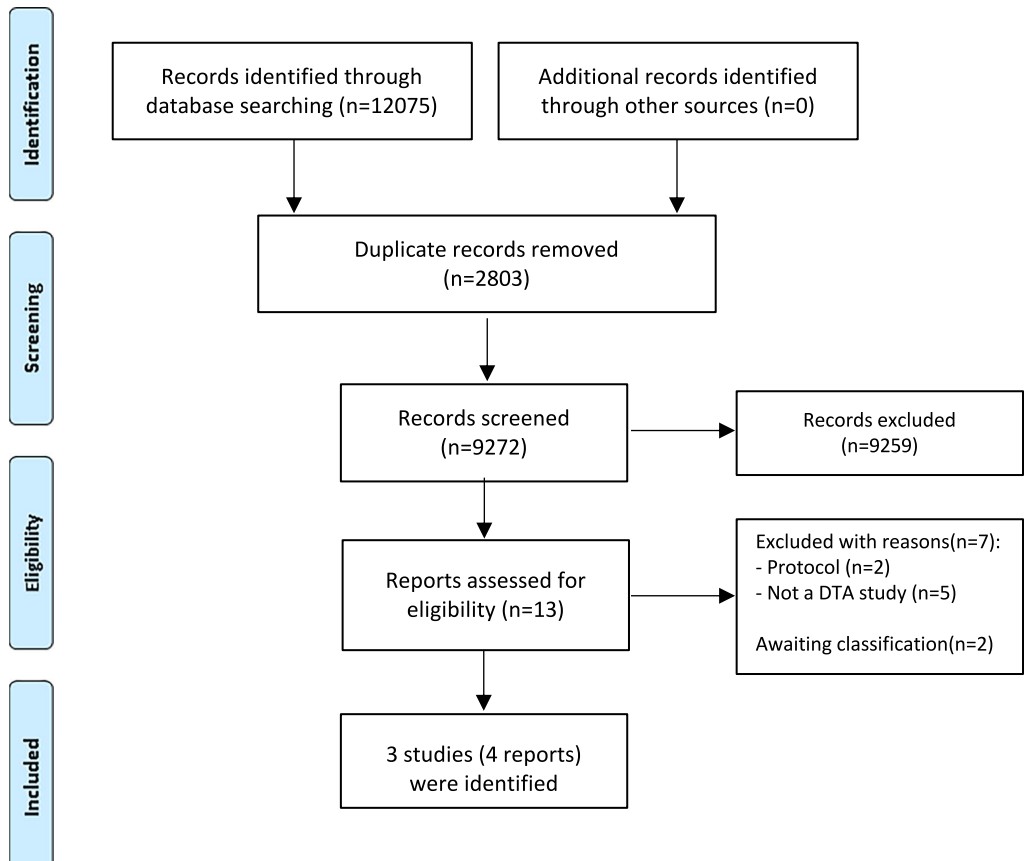

**Fig. 1 | PRISMA flow diagram.** Caption: This diagram illustrates how the included studies and reports were identified, starting from the database search. DTA–diagnostic test accuracy.

expense of an improvement in sensitivity (sensitivity 77% and specificity 67%)[34]. Finally, the addition of further laboratory investigations, available at a National referral hospital level, alongside echocardiography, led to a much better performing model (AUC 0.91, sensitivity 84% & specificity 87%) (Fig. 2).

Ali et al. assessed 400 febrile children prospectively in a cross-sectional hospital-based study conducted in a Pediatric Emergency Department in Al Obeid, North Kordofan state of Sudan[35]. The authors assessed the diagnostic yield of the modified 2015 Jones criteria without an echocardiogram and observed that 18 out of the 87 cases with definite ARF would be missed, as the only major criterion was subclinical carditis. Diagnostic accuracy of this approach revealed a drop in sensitivity, with preserved high specificity and discriminative capacity (sensitivity 79%, specificity 100% & AUC 0.90). Furthermore, 39 additional cases were considered possible ARF as they were missing only one minor criterion of evidence of streptococcal infection. A model considering both definite and possible ARF as positives displayed a drop in specificity with preserved high sensitivity and discriminative capacity (sensitivity 100%, specificity 88% & AUC 0.94).

Effects per 1000 patients with suspected ARF assessed with the different simplified diagnostic approaches are illustrated in Table 3.

Assessment of the performance of the diagnostic models in Goddard et al. was not possible as no info on the denominator (how many cases were assessed for potential ARF) was provided[37]. However, using data from the Northern Territories RHD register, cases of definite, probable, and possible ARF were identified. Probable ARF was defined as an acute presentation with ARF considered the most likely diagnosis, but not fulfilling Jones diagnostic criteria due to missing one major or one minor criterion or lacking evidence of preceding streptococcal infection. Possible ARF had a similar definition in terms of missing criteria, with ARF, despite being uncertain, could not be ruled out. Ten out of 181 probable

ARF cases (5.5%) had a subsequent ARF diagnosis during follow-up, and six out of 203 patients with possible ARF (3.0%) had a subsequent episode of probable or definite ARF.

### Development of RHD
Only two studies provided information on the development of RHD during follow-up. Goddard and colleagues reported increasing likelihood of progression based on ARF diagnosis probability on the initial event, with progression to RHD being reported in 5 out of 203 patients (2.5%) with possible ARF, 10 out of 181 patients (5.5%) with probable ARF, and 37 out of 348 patients (10.6%) with definitive ARF[37]. Pulle et al. followed 194 children from the study in Lira and Mbarara Regional Hospitals who did not meet modified Jones criteria for a definite ARF diagnosis for a period of >100 days[36]. On follow-up echocardiogram, three children had developed RHD (mild mitral regurgitation). Two of these children were part of a group of 80 (i.e., 2.5%) with a previous diagnosis of possible ARF, and the remaining child had an initial alternate diagnosis.

No data were available to conduct any of the pre-planned subgroup analyses.

### Discussion
We identified five different simplified algorithms for diagnosing ARF. These fall into two main categories: (i) algorithms with removal of some of the variables or investigations that are part of the 2015 Jones criteria, or (ii) adjustment of the threshold for positivity whilst using the 2015 Jones criteria (reducing the number of required minor or major criteria, and/or need for documenting streptococcal infection). These approaches require further testing, but do not seem to be a reliable alternative to the gold standard, the modified Jones criteria. To exemplify, removal of echocardiography, and consequently subclinical carditis as a major criterion, led to underdiagnosis.

## Table 1 | Characteristics of included studies and population

**Characteristics of included studies**

| Study | Country | Design | Sample | Study period | Setting | Patient selection | Follow-up period |
|---|---|---|---|---|---|---|---|
| Ali et al.[35] | Sudan | Cohort | 400 | September 2022–January 2023 | Pediatric Hospital in Al Obeid, North Kordofan | Children aged 3–18 presenting to pediatric ED with fever of unknown etiology ≤2 days in isolation or association signs/symptoms of ARF | NA |
| Goddard et al.[37] | Australia | Register | 913 | Jan 2013–June 2019 | Northern Territories | Indigenous Australians | 5.5 years |
| Ndagire et al.[33] Pulle et al.[36] | Uganda | Cohort | 503 | June 2017–June 2020 | Three districts: Lira, Kampala & Mbarara Three settings: Community health centers, District hospital & National referral hospital | Following a community sensitization session about signs and symptoms of ARF, participants were included in the study if they were aged 3–17 years, presented with fever and joint pain, or suspected to have carditis or chorea. | 461 days (167–820) |

**Patient characteristics**

| Study | Age | Female Sex n (%) | ARF n (%) | No ARF n (%) | Sore throat in past 4 weeks n (%) | Days of Fever median (IQR) | Tachycardia for age n (%) | Heart murmur n (%) | Family history ARF/ RHD n (%) |
|---|---|---|---|---|---|---|---|---|---|
| Ali et al.[35] | 3–18 | 180 (45) | 86 (21.5) | 314 (78.5) | 211 (52.8) | NA | NA | NA | NA |
| Goddard et al.[37] | 8–19 | 486 (53) | 509 (55.8) | NA | NA | NA | NA | NA | NA |
| Ndagire et al.[33] Pulle et al.[36] | 3–17 | 258 (51.3) | 143 (28.4) | 360 (71.6) | 146 (29.0) | 3 (1.6–5.0) | 150 (29.8) | 102 (20.3) | 13 (2.6) |

ARF acute rheumatic fever, IQR interquartile range, NA not available.

## Table 2 | Test details and performance

| Study | Simplified algorithm | TP | FP | FN | TN | Sens | Spec | LR+ | LR− | PPV | NPV | AUC | Reference Test |
|---|---|---|---|---|---|---|---|---|---|---|---|---|---|
| Ali et al.[35] | Jones criteria without echocardiogram | 69 | 0 | 18 | 313 | 79% (69–87) | 100% (99–100) | - | 0.21 | 100% | 99% | 0.90 | Modified Jones criteria 2015[8] |
| | Definite or Possible ARF | 87 | 39 | 0 | 274 | 100% (96–100) | 88% (83–91) | 8.03 | 0 | 2% | 100% | 0.94 | |
| Ndagire et al.[33] | Model 1 | Female sex, sore throat in past 4 weeks, heart murmur, family history of ARF/RHD, tachycardia, days of fever, medication for joint pain prior to visit, viral symptoms (rhinorrhea/cough), monoarthritis/polyarthritis/ polyarthralgia 91 | 113 | 47 | 239 | 66% (58–74) | 68% (63–73) | 2.05 | 0.50 | 37% | 88% | 0.69 | Modified Jones criteria 2015[8] |
| | Model 2 | Model 1 + PR interval ≥ 180 ms, WBC count, hemoglobin, confirmed malaria infection 105 | 114 | 31 | 233 | 77% (69–84) | 67% (62–72) | 2.35 | 0.34 | 40% | 91% | 0.76 | |
| | Model 3 | Model 2 + ESR, CRP, Streptococcal evidence, carditis on echocardiogram 114 | 44 | 21 | 298 | 84% (77–90) | 87% (83–90) | 6.56 | 0.18 | 65% | 95% | 0.91 | |

NA not available, TP true positives, FP false positives, TN true negatives, FN false negatives, Sens sensitivity, Spec specificity, LR+ positive likelihood ratio, LR− negative likelihood ratio, PPV positive predictive value, NPV negative predictive value, AUC area under receiver operating characteristic curve, ARF acute rheumatic fever, RHD rheumatic heart disease, WBC white blood cell, ESR erythrocyte sedimentation rate, CRP C-reactive protein.

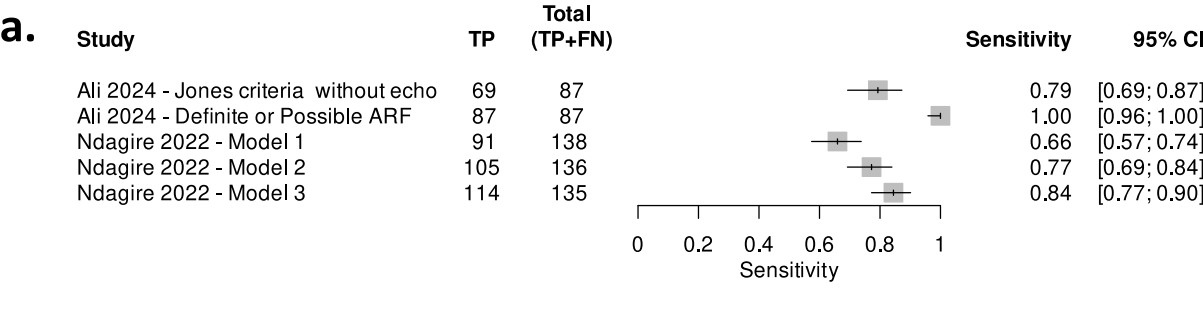

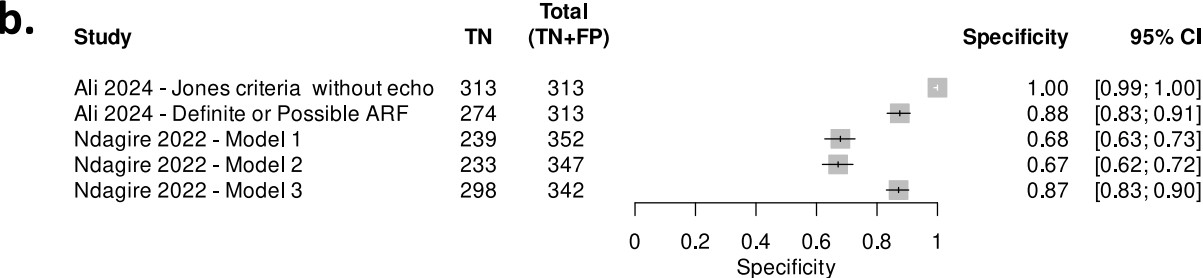

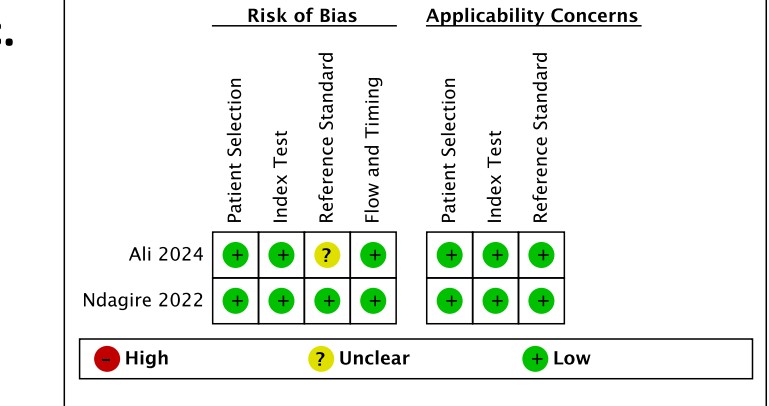

**Fig. 2 | Forestplots and Risk of Bias for diagnostic test accuracy studies. a, b** Show the performance of the different models for Sensitivity and Specificity, respectively. **c** Shows the risk of bias assessment for the two studies that provided information for these analyses. Note: "Unclear" for Ali 2024 as no antideoxyribonuclease-B antibody titers were available for some patients.

Sequentially adding investigations like 12-lead ECG, lab investigations, and echocardiography, to clinical information, in settings where these are available, leads to an improvement in diagnostic accuracy. Importantly, these data suggest that lower resource centers lacking some of the resources for applying the modified Jones criteria, and hence utilizing simplified diagnostic algorithms, may not be able to diagnose a relevant proportion of children presenting with ARF, delaying the initiation of antibiotics and preventing progression to RHD. On the other hand, accepting an ARF diagnosis in individuals who do not meet criteria for a definite ARF diagnosis may lead to overdiagnosis and false positives. A potential approach to overcome these issues is applying a two-stage diagnostic pathway: 1. Primary diagnosis of ARF in the primary healthcare setting with initiation of penicillin prophylaxis; 2. Subsequent referral to secondary care, with availability of echocardiography and other investigations for confirming or excluding the diagnosis[38].

Our results also suggest that the diagnosis of ARF remains challenging, and the modified 2015 Jones criteria are far from being a perfect clinical decision rule. This may be a result of the lack of laboratory and cardiac testing infrastructure in high-risk endemic areas, or due to the complex and progressive nature of ARF and potential lack of specificity of the modified Jones criteria in some cases. A previous study on Aboriginal Australian patients with suspected ARF showed that while a previous version of the Jones criteria could detect most definite ARF cases, vigilant follow-up was required to find the cases that initially did not fulfill the Jones criteria[39]. Diagnosis in mild episodes of ARF may be missed, with the patients missing out on timely prescriptions of secondary prophylaxis. This overlaps with the findings of the two studies included in our review, which followed patients who initially did not meet criteria for definite ARF. We observed that a small minority of those patients, 2–5% may end up being diagnosed with ARF and/or RHD. Ralph and colleagues have recommended supplementing the modified Jones criteria by actively excluding differential diagnoses and follow-up with echocardiography[39–41]. The Australian guideline recommends antibiotics for treatment of persisting streptococcal infection or asymptomatic respiratory tract carriage in individuals with definite, probable, or possible ARF[37]. This is similar to the recommendation of the 2015 modified Jones criteria scientific statement: in cases of "possible" ARF or uncertain diagnosis, *"it is reasonable to consider offering 12 months of secondary prophylaxis followed by reevaluation to include a careful history and physical examination in addition to a repeat echocardiogram (class IIa; level of evidence C)"*[8].

Identifying suspected cases earlier by applying broader clinical predictive rules (i.e. considering joint symptoms without an obvious alternative diagnosis), initiating antibiotics sooner and then referring for secondary care may be a pragmatic and effective approach, as suggested by Ali et al.[38].

**Table 3 | Effects per 1000 patients with suspected ARF assessed with the different simplified diagnostic approaches**

| Index test (no. of studies) | Reference standard | TP 95% CI | FN 95% CI | TN 95% CI | FP 95% CI |
|---|---|---|---|---|---|
| Modified Jones Criteria 2015 without echocardiogram | Modified Jones Criteria 2015 | 201 176–222 | 54 33–79 | 745 738–745 | 0 0–7 |
| Definite or Possible ARF | Modified Jones Criteria 2015 | 255 245–255 | 0 0–10 | 656 618–678 | 89 67–127 |
| Model 1 | Modified Jones Criteria 2015 | 168 148–189 | 87 66–107 | 507 469–544 | 238 201–276 |
| Model 2 | Modified Jones Criteria 2015 | 196 176–214 | 59 41–79 | 499 462–536 | 246 209–283 |
| Model 3 | Modified Jones Criteria 2015 | 214 196–230 | 41 25–59 | 648 618–671 | 97 74–127 |

Based on observed prevalence of ARF of 25.5% among children evaluated for suspected disease in the two studies.

This is particularly important in low-resource settings, where patients may face delays of weeks to months before receiving secondary care. Such delays can postpone the initiation of antibiotics, increasing the risk of complications, including potential cardiac sequelae.

Our systematic review indicates that simplifying the diagnostic criteria for ARF beyond the current modified Jones criteria for broader clinical application may not be feasible in the near term with the currently available array of investigations. Ongoing research for a better understanding of the biomarker signature in ARF and RHD[19,42] could yield additional diagnostic instruments that enhance early detection accuracy for both probable and possible ARF cases. Currently, no single biomarker for ARF is both highly sensitive and specific. C-reactive protein and erythrocyte sedimentation rate have low specificity, as they can be elevated in various inflammatory or infectious conditions. Anti-streptolysin O and anti-DNase B titers have moderate sensitivity—meaning they may not always be elevated if a streptococcal infection occurred weeks before ARF symptoms—and limited specificity, as they indicate prior Group A Streptococcus infection but do not confirm ARF. Five proteins, identified using *Somalogic's SomaScan®* proteomic assay, appear to offer high sensitivity and specificity when combined for diagnosing ARF[43]. However, further studies, with validation in external cohorts, and development of an assay for use in clinical practice are still required before this promising approach can be broadly implemented.

RHDAustralia has addressed the intricacies of the modified 2015 Jones criteria and their practical application challenges by introducing a digital solution. The ARF RHD app, available in app stores, encapsulates the criteria and has performed well in its evaluation[44], and garnered favorable reviews for its utility over traditional paper-based algorithms, indicating a preference among clinicians for this digital tool.

Enhancing diagnostic accuracy for ARF may involve comprehensive exclusion of alternative diagnoses and consideration of symptoms not classified as major or minor criteria, such as chest pain indicative of pericarditis, arrhythmias, and NSAID responsiveness. These methods, while thorough, add complexity to the diagnostic process rather than simplifying it.

The application of the modified Jones criteria is currently beyond the capacity of most community and district-level health facilities in regions with high ARF incidence. The absence of essential diagnostic services, including laboratory tests and echocardiography, is a barrier to the improvement of the diagnostic accuracy of ARF in those areas. Importantly, expanding the use of antibiotic treatment to suspected cases of streptococcal pharyngitis in populations at moderate to high-risk of ARF and RHD appears to be of importance for reducing the burden of ARF[45]. Oral or intramuscular penicillin treatment of streptococcal pharyngitis has been shown to reduce ARF within 2 months by 64%[46], suggesting a role for primary prevention of ARF with antibiotics.

Finally, assessing the cost-effectiveness of interventions in high-prevalence areas is of utmost importance. An economic model estimated the incremental cost-effectiveness of scaling up coverage of primary and secondary prevention and heart valve surgery interventions for RHD. Its preliminary findings suggested that primary prevention (i.e., timely diagnosis and adequate treatment of the superficial group A Streptococcal infections, pharyngitis, and impetigo) is the most efficient and cheapest approach in low-resource countries[47]. The two primary prevention strategies that appear to be particularly effective are school-based clinics to diagnose and treat group A Streptococcal pharyngitis, and using antibiotics in children with a positive group A Streptococcal throat swab[48].

The financial implications and feasibility of integrating new biomarker tests and broader imaging procedures must be carefully weighed, as they may exacerbate the economic strain on healthcare systems in such regions.

The following unresolved research questions and areas of future research were identified by this systematic review to inform the new World Health Organization guideline on ARF and RHD:

- There is a variable capacity to apply the modified Jones criteria in a highly endemic area of ARF. Improvement of ARF diagnostic accuracy may face challenges due to a lack of availability and access to laboratory investigations and echocardiography in such areas.
- Further research is required on more specific biomarkers to diagnose ARF or that can act as supplementary diagnostic criteria to modify the Jones criteria to increase the criteria's specificity and sensitivity.
- More data are needed on the possibility of categorizing suspected ARF cases into categories based on diagnostic certainty and following a management pathway specific to each category.
- The development of multiparametric risk scores to bridge this knowledge gap should conform to established methodologies, such as the TRIPOD framework[49,50].
- The meticulous exclusion of differential diagnoses remains crucial for refining the accuracy of the modified Jones criteria.
- Use of diagnostic aids, such as apps, to make the criteria more user-friendly and easier to use, and facilitating the use of more elaborate criteria needs to be tested. This approach can potentially pave the way for more complex decision flowcharts, shifting away the focus from criteria simplification.

The scarcity of studies meeting the inclusion criteria is a limitation of this review that needs to be highlighted. Due to the presence of different algorithms, each of which was only used in a single study, we were unable to conduct a meta-analysis. A future update of this review may allow circumvent some of the uncertainty and limitations in our findings. Furthermore, reporting of data per age group should be encouraged in future publications, so sub-analyses may become feasible.

In conclusion, ARF remains frequently underdiagnosed, highlighting the need for a simple and broadly applicable approach that can detect the vast majority of cases. Although simplified algorithms that omit echocardiography and laboratory testing have shown suboptimal performance, even the modified Jones criteria have limitations (some patients who do not meet the criteria for a definite ARF diagnosis may still progress to RHD). This highlights a critical gap in current diagnostic methods. Further research is needed to identify reliable clinical and/or biochemical markers for ARF, particularly in endemic regions, to improve diagnostic accuracy and prevent progression to RHD. Additionally, the use of antibiotics in populations at moderate to high risk of ARF and RHD with suspected streptococcal pharyngitis may be of importance for reducing the burden of ARF.

## Data availability

All data used for the analyses were extracted from published studies and are included in the Article. Additional details on Ndagire and colleagues[34] were

provided by the study authors. No patient-level data were used. Additional information is available via contact with the corresponding author.

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

## Acknowledgements

Dr. Erik von Elm (Cochrane Switzerland, Unisanté Lausanne) & Dr Susan L. Norris (WHO Guideline writer) independently peer-reviewed this review's protocol and final report. We thank Professor Anna Ralph (Menzies School of Health Research) for generously sharing her expert knowledge and information about their study that helped us decide on its eligibility. Dr. Emma Ndagire, Dr. Andrea Beaton, and Dr. Nicholas J. Ollberding for providing additional data from their study for the purpose of this systematic review. The World Health Organization (WHO) commissioned this review to support the development of WHO's clinical practice guidelines on the prevention and management of Acute Rheumatic Fever and Rheumatic Heart Disease. Review Question 9: "*Among children, adolescents and adults with suspected RF what is the diagnostic accuracy of simplified algorithms (as compared to modified Jones criteria) to identify RF at primary health facilities?*" The funder supplied the research questions, defined the PICO (Population, Index test, Comparators, Outcomes), and had no role in study design, collection, analysis, and interpretation of data, or writing of the report. The funder commissioned independent reviewers who commented on the review's protocol and final report several times. As part of the commission, we agreed to submit the resulting output to a peer-reviewed journal.

## Author contributions

R.P. and G.A. wrote the final draft of the manuscript. F.Z., T.K. provided methods input. F.S. and F.P. provided information specialist expertise. M.A., J.J.H.B., E.M., M.C., and D.C. provided clinical input. All authors revised the first draft of the manuscript and provided comments to improve it and prepare the final version. All authors read and approved the final version of the manuscript.

## Inclusion and ethics statement

All authors met the authorship criteria, and their roles and responsibilities were agreed upon prior to the research. Findings were discussed with the WHO development group which assured these were representative of the regional reality in high-prevalence areas of ARF and RHD.

## Competing interests

The authors declare no competing interests.
