## [Transparent Peer Review file · Communications Medicine]

Diagnostic Test Accuracy of Simplified Algorithms for Diagnosing Acute Rheumatic Fever: A Systematic Review

Corresponding Author: Professor Rui Providencia

Version 0:

Reviewer comments:

Reviewer #1

(Remarks to the Author)

1. Authors do a nice job of setting the epidemiological basis for the study as well as the importance on a societal level due to medical costs
2. Lines 102-105, it would be nice to further define the genetic susceptibility as well as the validity (or lack thereof) of the mentioned biomarkers in disease diagnosis and treatment and what has been published.
3. The authors do a nice job of outlining their relatively rigorous and thorough methods in determining eligible studies and assessment of these studies. Also the subgroup analyses seem very appropriate, although it is unfortunate the data was not available for this (by no fault of the authors). Figure 2 and Table 1 are particularly helpful for the reader to conceptualize this framework.
4. Authors do a nice job of highlighting the need for formalized studies on the efficacy of modified Jones criteria, particularly in countries where Group A Strep is more prevalent and access to resources is more limited as well as emphasizes the need for biomarkers. Here again it may be helpful to spend time discussing what is known in the literature about this, and/or emphasize the limited amount of biomarkers available for this specific disease process (Lines 304-308).

Reviewer #2

(Remarks to the Author)

This is a very important and highly needed area of research. The study is well planned, with excellent methodology, results and discussion sections. There are no major concerns regarding any of the sections.

I have few suggestions:

1.Line 75 (Background):

There is no evidence that RHD causes structural myoacrdial disease

2.Line 110: authors need to highlight more of the Jones Criterial limitations including the large number of items, need for lab and the inherent nature of ARF where arthritis is transient and carditis subclinal.

3. Line 286: one of the suggested methods to overcome delayed diagnosis of ARFT is applying a 2-stage diagnostic pathway: one is primary diagnosis of ARF at primary health care setting, and initiating BPG prophylaxis, then referring to secondary care level where echo/other investigations are done and diagnosis either confirmed or ruled out.

Ref:Ali SKM. Rheumatic heart disease control: the time for a paradigm shift. Sudan J Paediatr. 2022;22(2):125-130. doi: 10.24911/SJP.106-1652814717. PMID: 36875951; PMCID: PMC9983771.

4.Line 303: authors could include the comment ARF should be diagnosed at primary health care level to pick up suspected cases earlier by applying broader clinical predictive rule such as joint symptoms without obvious alternative diagnosis, start antibx, refer for secondary care.in order to start prophylaxis early: most patients cannot go to the next care level except after

weeks-months therefore prophylaxis needs to be considered at that point and continued till next visit.

5. Line 352:

However, a meta-analysis did not show significant benefit for primary prophylaxis

Bennett J, Rentta N, Leung W, Anderson A, Oliver J, Wyber R, Harwod M, Webb R, Malcom J, Baker MG. Structured review of primary interventions to reduce group A streptococcal infections, acute rheumatic fever and rheumatic heart disease. *J Paediatr Child Health*. 2021 Jun;57(6):797-802. doi: 10.1111/jpc.15514. Epub 2021 Apr 20. Erratum in: *J*

Reviewer #3

(Remarks to the Author)

Dear authors

Thank you for your submission,

The main problem with the study is that you conclude that simplified criteria would lead to underdiagnosis of RF. The easier and less variables in the criteria, the more diagnosis will be actually made. Plus, echocardiography will not help in the diagnosis of RF unless there is carditis, which is not always the fact.

We are aiming for simplification of all criteria (echocardiography included) to increase the rate of RF diagnosis, as many RHD patients do not report a diagnosis of RF, which is possibly the result of underdiagnosis, which leads to failing antibiotic prophylaxis to prevent RHD.

Furthermore,

Small Number of Included Studies:

The review includes only three studies, which limits the generalizability of the findings. The authors acknowledge this limitation, but it remains a significant constraint. Expanding the search to include more studies or providing a more detailed discussion of how this limitation affects the results would strengthen the manuscript.

Lack of Robust Statistical Analysis:

The meta-analysis is limited by the small number of studies, and the results show significant heterogeneity. The authors could consider conducting a more robust statistical analysis, such as subgroup analysis or sensitivity analysis, to explore the sources of heterogeneity and provide more nuanced insights.

Best regards,

Susy

Version 1:

Reviewer comments:

Reviewer #1

(Remarks to the Author)

The authors did an excellent job of incorporating feedback into the article.

Reviewer #2

(Remarks to the Author)

This statement in conclusion:

Simplification of the modified Jones criteria in areas without access to echocardiography and laboratory investigations may lead to under-diagnosis of ARF.

does not reflect the essence of this review.

I suggest to mention facts such as,

It is highly needed to have a simple, sensitive approach to diagnose ARF as it has been proven that ARF is often underdiagnosed, however, current evidence does not support the published algorithms.

More research is needed to identify clinical and/or biochemical diagnostic markers for ARF in endemic areas in order to

prevent RHD.

Reviewer #3

(Remarks to the Author)

Thank you for the revisions

Reviewer #1 (Remarks to the Author):

1. Authors do a nice job of setting the epidemiological basis for the study as well as the importance on a societal level due to medical costs

Thanks for this this remark.

2. Lines 102-105, it would be nice to further define the genetic susceptibility as well as the validity (or lack thereof) of the mentioned biomarkers in disease diagnosis and treatment and what has been published.

Thank you for raising this important aspect. We have now added further information about this topic.

Background:

“Advances in understanding the pathophysiology of ARF suggest that genetic susceptibility may play a role¹³⁻¹⁵. Previous analyses of twin studies showed a concordance rate of 44% in monozygotic twins and 12% in dizygotic twins, with the heritability of ARF estimated to be 60%¹⁴. Genetic variants within the human leukocyte antigen (HLA) system and other immune processes have been identified¹³. Furthermore, the interest in biomarkers (i.e. Periostin, Tenascin-C, ischemia-modified albumin, Interferon- γ induced protein 10, IgG3-C4, etc.) has been growing¹⁶⁻²⁰, but convincing data on their validity for replacing the Jones criteria or to help improve their diagnostic accuracy is currently lacking.”

3. The authors do a nice job of outlining their relatively rigorous and thorough methods in determining eligible studies and assessment of these studies. Also the subgroup analyses seem very appropriate, although it is unfortunate the data was not available for this (by no fault of the authors). Figure 2 and Table 1 are particularly helpful for the reader to conceptualize this framework.

Thanks for noting this. Hopefully, in a few years' time, there will be more studies and such sub-analyses may be possible. In the discussion section we reinforce the need of more detailed reporting in future publications so that sub-analyses may be made possible:

Discussion:

“Furthermore, reporting of data per age group should be encouraged in future publications, so sub-analyses may become feasible.”

4. Authors do a nice job of highlighting the need for formalized studies on the efficacy of modified Jones criteria, particularly in countries where Group A Strep is more prevalent and access to resources is more limited as well as emphasizes the need for biomarkers. Here again it may be helpful to spend time discussing what is known in the literature about this, and/or emphasize the limited amount of biomarkers available for this specific disease process (Lines 304-308).

Thanks for this suggestion. We have expanded this section.

Discussion:

“Currently, no single biomarker for acute rheumatic fever (ARF) is both highly sensitive and specific. C-reactive protein (CRP) and erythrocyte sedimentation rate (ESR) have low specificity, as they can be elevated in various inflammatory or infectious conditions. Anti-streptolysin O (ASO) and anti-DNase B titers have moderate sensitivity—meaning they may not always be elevated if a streptococcal infection occurred weeks before ARF symptoms—and limited specificity, as they indicate prior Group A Streptococcus infection but do not confirm ARF. Five proteins, identified using Somalogic’s SomaScan® proteomic assay, appear to offer high sensitivity and specificity when combined for diagnosing ARF⁴⁴. However, further studies, with validation in external cohorts, and development of an assay for use in clinical practice are still necessary before this promising approach can be broadly implemented.”

Reviewer #2 (Remarks to the Author):

This is a very important and highly needed area of research. The study is well planned, with excellent methodology, results and discussion sections. There are no major concerns regarding any of the sections.

I have few suggestions:

1.Line 75 (Background):

There is no evidence that RHD causes structural myocardial disease

Thanks for noticing this imprecision. The mention of myocardial involvement has now been removed.

“Rheumatic Heart Disease (RHD) is a condition characterized by structural and functional damage to the heart valves as a result of ARF.”

2.Line 110: authors need to highlight more of the Jones Criteria limitations including the large number of items, need for lab and the inherent nature of ARF where arthritis is transient and carditis subclinical.

Thanks for this suggestion. We have now provided further detail to the Background section.

“Importantly, these criteria are complex, comprising multiple components, and often require laboratory tests and/or echocardiography, which may not always be accessible in low- and middle-income countries.”

“Current modified Jones criteria rely heavily on clinical manifestations, which can be subjective, transient or hard to detect (i.e. arthritis is transient and carditis may be sub-clinical) and lead to over and underdiagnosis.”

3. Line 286: one of the suggested methods to overcome delayed diagnosis of ARFT is applying a 2-stage diagnostic pathway: one is primary diagnosis of ARF at primary health care setting, and initiating BPG prophylaxis, then referring to secondary care level where echo/other investigations are done and diagnosis either confirmed or ruled out.

Ref:Ali SKM. Rheumatic heart disease control: the time for a paradigm shift. Sudan J Paediatr. 2022;22(2):125-130. doi: 10.24911/SJP.106-1652814717. PMID: 36875951; PMCID: PMC9983771.

Thanks for suggesting this important workflow. We have now added this information to the discussion section.

“A potential approach to overcome these issues is applying a two-stage diagnostic pathway: 1. Primary diagnosis of ARF in the primary health care setting with initiation of penicillin prophylaxis; 2. Subsequent referral to secondary care, with availability of echocardiography and other investigations for confirming or excluding the diagnosis³⁸.”

4.Line 303: authors could include the comment ARF should be diagnosed at primary health care level to pick up suspected cases earlier by applying broader clinical predictive rule such as joint symptoms without obvious alternative diagnosis, start antibx, refer for secondary care.in order to start prophylaxis early: most patients cannot go to the next care level except after weeks-months therefore prophylaxis needs to be considered at that point and continued till next visit.

This is an important remark. We have now added this to the revised Discussion section.

“Identifying suspected cases earlier by applying broader clinical predictive rules (i.e. considering joint symptoms without an obvious alternative diagnosis), initiating antibiotics sooner and then referring for secondary care may be a pragmatic and effective approach, as suggested by Ali et al³⁸. This is particularly important in low-resource settings, where patients may face delays of weeks to months before receiving secondary care. Such delays can postpone the initiation of antibiotics, increasing the risk of complications, including potential cardiac sequelae.”

5. Line 352:

However, a meta-analysis did not show significant benefit for primary prophylaxis

Bennett J, Rentta N, Leung W, Anderson A, Oliver J, Wyber R, Harwod M, Webb R, Malcom J, Baker MG. Structured review of primary interventions to reduce group A streptococcal infections, acute rheumatic fever and rheumatic heart disease. J Paediatr Child Health. 2021 Jun;57(6):797-802. doi:

10.1111/jpc.15514. Epub 2021 Apr 20. Erratum in: J

We have now incorporated this reference and highlighted the two strategies that Bennett and colleagues have identified as the most effective for primary prevention.

Discussion:

“The two primary prevention strategies that appear to be particularly effective are school-based clinics to diagnose and treat group A Streptococcal pharyngitis, and using antibiotics in children with a positive group A Streptococcal throat swab⁴⁹.”

Reviewer #3 (Remarks to the Author):

The main problem with the study is that you conclude that simplified criteria would lead to underdiagnosis of RF. The easier and less variables in the criteria, the more diagnosis will be actually made. Plus, echocardiography will not help in the diagnosis of RF unless there is carditis, which is not always the fact.

We thank the Reviewer for this comment. As we show in the manuscript, among the tested algorithms, dropping some of the modified Jones criteria would result in missing a few diagnoses of ARF. We agree with the desirability of developing easier criteria, with fewer variables, but such criteria would have to have equal diagnostic performance to be implemented safely. We add further to this important remark by the reviewer in the next point.

We are aiming for simplification of all criteria (echocardiography included) to increase the rate of RF diagnosis, as many RHD patients do not report a diagnosis of RF, which is possibly the result of underdiagnosis, which leads to failing antibiotic prophylaxis to prevent RHD.

This is a great remark. As the Reviewer suggests, not only the simplification of the diagnostic criteria for ARF, but also the expansion of antibiotic treatment of suspected cases of streptococcal pharyngitis in populations at moderate to high risk of ARF and RHD are important goals for reducing the global burden of ARF (ref WHO guideline). Oral or intramuscular penicillin have been shown to reduce ARF within 2 months by 64% (ref Spinks et al.).

Discussion:

“Importantly, expanding the use of antibiotic treatment to suspected cases of streptococcal pharyngitis in populations at moderate to high-risk of ARF and RHD appears to be of importance for reducing the burden of ARF ⁴⁶. Oral or intramuscular penicillin treatment of streptococcal pharyngitis have been shown to reduce ARF within 2 months by 64% ⁴⁷, suggesting a role for primary prevention of ARF with antibiotics.”

References:

World Health Organization. WHO guideline on the prevention and diagnosis of rheumatic fever and rheumatic heart disease. Geneva: World Health Organization; 2024. Available at: <https://www.who.int/publications/i/item/9789240100077>

**Furthermore,
Small Number of Included Studies:**

The review includes only three studies, which limits the generalizability of the findings. The authors acknowledge this limitation, but it remains a significant constraint. Expanding the search to include more studies or providing a more detailed discussion of how this limitation affects the results would strengthen the manuscript.

We thank the Reviewer for this suggestion. We have now updated the search until March 15th 2025. Unfortunately, no further studies for inclusion were identified. There are two studies of potential interest, but data available in the abstract are not enough to make a final decision on inclusion. We highlight this matter at the end of the discussion section:

Results:

“Our broad search identified 12,075 records, and we screened 9,272 unique records. We found ten records possibly relevant during the record screening phase, but seven were excluded after more detailed assessment, and two were abstracts where further data is required for making a final decision (Figure 1).”

Extract of Table S-1 with the two studies awaiting classification:

Okello et al. 2024	Awaiting Classification. No data available for DTA. Abstract only. Once published as a manuscript may need to be rescreened. Describes a cohort of 165 children, 50 with definite ARF and the remaining with other clinically overlapping conditions (no details provided). Two sites involved: central vs western recruitment site (not details provided on location). Utilizing SomaScan proteomic profiling (no details provided on version) twenty-two proteins significantly associated with ARF were identified, and 5 had very good discriminative capacity for clinical cases (not specified) when combined: central training cohort (AUC 0.97, 95% CI 0.90-0.97), and western testing cohort (AUC 0.84, 95% CI 0.71-0.84). These proteins returned to baseline on convalescent phase. No uniprot id or protein names were provided.
Panwar et al. 2025	Awaiting Classification. No data available for DTA. Abstract only. Once published as a manuscript may need to be rescreened. Describes a cohort of 200 Children with febrile illness and sore throat but lacking ARF features. Screened with echocardiogram; the abstract mentions 7 cases (3.5%) of RHD cases were discovered. However, no data on modified jones criteria and DTA with and without echocardiography is presented.

Discussion:

“The scarcity of studies meeting the inclusion criteria is a limitation of this review that needs to be highlighted. Due to the presence of different algorithms, each of which was only used in a single study, we were not able to perform a meta-analysis. A future update of this review may allow to circumvent some of the uncertainty and limitations in our findings. Furthermore, reporting of data per age group should be encouraged in future publications, so sub-analyses may become feasible.”

Lack of Robust Statistical Analysis:

The meta-analysis is limited by the small number of studies, and the results show significant heterogeneity. The authors could consider conducting a more robust statistical analysis, such as subgroup analysis or sensitivity analysis, to explore the sources of heterogeneity and provide more nuanced insights.

We thank the Reviewer for this suggestion. Unfortunately, as mentioned in the manuscript we could not perform any of the pre-planned sub-analyses as the data were not available. Furthermore, due to the existence of various algorithms, each of which was only used in a single study, we were unable to conduct a meta-analysis. As such, a sensitivity analysis removing potential outliers was not feasible. We opted for presenting sensitivity and specificity results in a plot and a narrative description of findings.

Discussion:

“Due to the presence of different algorithms, each of which was only used in a single study, we were unable to conduct a meta-analysis.”

Reviewer #2 (Remarks to the Author):

This statement in conclusion:

Simplification of the modified Jones criteria in areas without access to echocardiography and laboratory investigations may lead to under-diagnosis of ARF.

does not reflect the essence of this review.

I suggest to mention facts such as,

It is highly needed to have a simple, sensitive approach to diagnose ARF as it has been proven that ARF is often underdiagnosed, however, current evidence does not support the published algorithms.

More research is needed to identify clinical and/or biochemical diagnostic markers for ARF in endemic areas in order to prevent RHD.

We thank the Reviewer for the Suggestion. We have incorporated its message into the revised Conclusion:

“In conclusion, ARF remains frequently underdiagnosed, highlighting the need for a simple and broadly applicable approach that can detect the vast majority of cases. Although simplified algorithms that omit echocardiography and laboratory testing have shown suboptimal performance, even the modified Jones criteria have limitations (some patients who do not meet the criteria for a definite ARF diagnosis may still progress to RHD). This highlights a critical gap in current diagnostic methods. Further research is needed to identify reliable clinical and/or biochemical markers for ARF, particularly in endemic regions, to improve diagnostic accuracy and prevent progression to RHD. Additionally, the use of antibiotics in populations at moderate to high-risk of ARF and RHD with suspected streptococcal pharyngitis may be of importance for reducing the burden of ARF.”